# Patient Safety Culture of Hospitals in Southern Laos: A Cross-Sectional Study Using the Hospital Survey on Patient Safety Culture

**DOI:** 10.3390/healthcare13151934

**Published:** 2025-08-07

**Authors:** Miho Sodeno, Moe Moe Thandar, Somchanh Thounsavath, Olaphim Phouthavong, Masahiko Hachiya, Yasunori Ichimura

**Affiliations:** 1Japan Institute for Health Security, Bureau of Global Health Cooperation, Tokyo 162-8655, Japanichimura.y@jihs.go.jp (Y.I.); 2Global Health and Research Center, Department of Medicine, Juntendo University, Tokyo 113-8421, Japan; 3Department of Healthcare and Rehabilitation, Ministry of Health, Vientiane 01030, Laos; jecthcd@yahoo.com (S.T.); olaphim@hotmail.com (O.P.)

**Keywords:** patient safety, safety management, hospitals, surveys and questionnaires, Laos

## Abstract

**Background:** Patient safety culture is critical for enhancing the quality and safety of healthcare. Studies in low- and middle-income countries have reported challenges in developing patient safety culture, especially in implementing nonpunitive responses to errors and event reporting. However, evidence from Laos remains limited. **Objectives:** This study aimed to assess patient safety culture in hospitals in southern Laos, using a validated survey tool to identify strengths and areas of improvement. **Methods**: A cross-sectional study using purposive sampling was conducted in four provincial and twenty-three district hospitals in southern Laos. Healthcare workers on patient safety committees responded to the Hospital Survey on Patient Safety Culture. The positive response rate was analyzed. Bivariate tests (chi-square/Fisher’s exact) were applied to compare positive response rates between hospital types and professions. **Results**: A total of 253 valid responses (75.5%) were analyzed. “Organizational Learning–Continuous Improvement” scored over 75% in both provincial and district hospitals. In contrast, “Nonpunitive Response to Error” and “Frequency of Events Reported” were scored <20% on average. Provincial hospitals scored significantly higher than district hospitals in supervisory support and handoffs. **Conclusions**: This study illustrated strengths in organizational learning while identifying nonpunitive responses and event reporting as critical areas of improvement for hospitals in Laos. To improve patient safety, hospitals in Laos must promote a culture in which errors can be reported without fear of blame. Strengthening leadership support and reporting systems is essential. These findings can inform strategies to enhance patient safety in other low-resource healthcare settings.

## 1. Introduction

Patient safety is a critical issue in healthcare systems, and patient safety culture (PSC) is a key factor in improving healthcare quality and protecting patients globally [1]. PSC refers to the shared values, beliefs, and norms about the importance of patient safety [2]. A strong PSC has been associated with improved quality of care and reduced adverse events [3,4,5]. Strengthening PSC is particularly essential for health systems aiming to provide safe and effective care, as it enhances service delivery, protects patients from harm, supports staff morale, and fosters public trust in the healthcare system. Interventions promoting PSC can improve patients’ perceptions of healthcare workers and potentially reduce patient harm [6,7,8]. The importance of PSC has been stressed by the World Health Organization, which included it as a central goal in its Global Patient Safety Action Plan for 2021–2030 [9].

One of the most widely used tools for assessing PSC is the Hospital Survey on Patient Safety Culture (HSOPSC), developed by the United States Agency for Healthcare Research and Quality (AHRQ). This tool assesses many dimensions of PSC, such as teamwork, communication, error reporting, and the overall approach to learning from mistakes. As of September 2022, the HSOPSC 1.0 has been validated in multiple languages and applied in more than 90 countries [10,11].

In Southeast Asia, the state of PSC varies across countries, and implementation is often lower compared with that in high-income settings [12,13]. For example, in Vietnam, the use of the HSOPSC has identified “Nonpunitive Response to Error” as an area needing improvement [14]. However, in the Lao People’s Democratic Republic, or Laos, evidence on patient safety remains extremely limited [12].

Laos is a lower-middle-income country with a population of approximately 7.5 million and demonstrates a relatively sparse population density compared with its Southeast Asian neighbors [15]. The healthcare system in Laos primarily comprises public facilities and is structured into four administrative levels: central, provincial, district, and health centers [16]. The Ministry of Health oversees national policy and planning, while provincial and district health offices are responsible for supervising the hospitals and health centers within their jurisdictions [15,16,17]. Although Laos has implemented a national health insurance scheme aimed at achieving universal health coverage, challenges remain in equitable access and resource distribution, particularly in remote areas [16]. Public hospitals are the main providers of inpatient and outpatient services, and the referral system follows the hierarchical structure of the healthcare administration [16,17].

Many hospitals in Laos experience shortages of trained healthcare workers and limited training opportunities, which hinders the establishment of a safe care environment [15]. The country currently lacks national guidelines focused on patient safety. Furthermore, to the best of our knowledge, no previous study has applied standardized tools to assess PSC in Laos. This gap limits the ability to identify priority areas for improvement or provide targeted support to healthcare workers. A clear understanding of the current state of PSC is essential for designing effective, context-specific interventions and policies.

This study aimed to assess patient safety culture in hospitals in southern Laos using a validated survey tool to identify strengths and areas of improvement.

## 2. Methods

### 2.1. Study Design

A hospital-based cross-sectional study was conducted in the four provinces of southern Laos (Attapeu, Sekong, Salavan, and Champasak). These provinces host 4 public provincial hospitals (PHs) and 23 public district hospitals (DHs). Each province has one PH and three to nine DHs [18]. These hospitals were also part of the Japan International Cooperation Agency (JICA) Quality Health Care for the Future project, which addressed healthcare quality and financial management from March 2022 until March 2025 [19]. At the time of the study, patient safety committees had been established in the target hospitals across the four provinces but had not yet commenced their activities [20].

### 2.2. Participants and Sampling

We used purposive sampling to recruit healthcare workers serving on patient safety committees at 27 public hospitals (4 PHs and 23 DHs) across four provinces of Laos. Based on preliminary information on the number of patient safety committee members at each hospital, the planned sample size assumed an 80% response rate. Eligible participants were those aged 20 years or older who provided informed consent. Participants were selected to represent various professional roles related to patient safety (e.g., physicians, nurses, pharmacists). No exclusion criteria were applied based on professional experience, as each hospital’s management selected a diverse committee. Recruitment numbers per hospital were based on facility size: 25–30 participants were recruited from each PH and 10 from each DH.

### 2.3. Questionnaire

This study used the HSOPSC version 1.0 [2], which contained 42 items across the 12 dimensions, as shown in Table 1. Responses were rated on a five-point Likert scale, ranging from “Strongly Disagree” to “Strongly Agree” for agreement items and from “Never” to “Always” for frequency-based items. The HSOPSC questionnaire was translated into Lao by a native Lao speaker with proficiency in English and familiarity with healthcare terminology.

To ensure accuracy and consistency, the translated version was back-translated into English by another native Lao speaker who was proficient in English and not involved in the initial translation process. A preliminary comprehension check was conducted with Lao-speaking physicians and nurses to confirm the clarity and contextual relevance of the items. Additionally, feedback-based review was conducted by physicians in the Department of Healthcare and Rehabilitation in the four provincial health offices. Based on their feedback, brief explanatory notes were added to the questionnaire. The term “culture” lacked an appropriate equivalent in Lao; it is often interpreted as “history,” which could have caused misunderstanding. Therefore, we provided a brief explanation of the term using the HSOPSC definition: “PSC refers to the shared values, beliefs, and norms about the importance of patient safety within an organization.”

### 2.4. Data Collection

The questionnaire was distributed on paper, and data was collected from October 2023 to January 2024. The research team explained the study objectives and data collection procedures to the directors of the PHs and DHs and the officials from the provincial health offices overseeing these hospitals. The team then distributed the information documents detailing the study’s purpose and procedures, along with the questionnaires, to the public health officials of each province. They then distributed these documents to the head of the patient safety committee of each hospital, who in turn distributed them to the patient safety committee members. The participants used their free time to answer on their own, completed the questionnaires, and independently returned them to the drop-off box in each facility. The head of the patient safety committee provided them to the public health officials in sealed envelopes. The public health officials collected and sent them to the research team. All participants were explicitly informed, both in writing and verbally, that their responses would remain anonymous. Following HSOPSC guidelines, which recommend withholding facility-level results for hospitals with fewer than 10 respondents to preserve confidentiality, participants in DHs were also informed that hospital-specific results would not be disclosed.

### 2.5. Analysis

The research team followed the HSOPSC User’s Guide to analyze the survey data [2]. The percentage of positive responses for each survey item was calculated, with positive responses being defined according to the HSOPSC guidelines. These percentages were then averaged to determine the scores for each dimension. For items using a five-point Likert scale, responses of “Agree,” “Strongly Agree,” “Most of the Time,” or “Always” were classified as positive responses. Responses like “Strongly Disagree” or “Never” were similarly categorized as negative responses for descriptive purposes but were not included in the scoring process. Negatively worded items were reverse-coded as per the AHRQ guidelines. The mean positive response rate (PRR) of each item of a dimension was calculated at the individual level. For each participant, the dimensions’ scores were derived by averaging the responses to the relevant items and converting the mean to a percentage, the mean PRR. To ensure accuracy, participants with missing responses for any items within a dimension were excluded from these calculations. In the “Teamwork Within Units” dimension, one item was excluded due to a misalignment in the questionnaire version during adaptation. Specifically, a conceptually unrelated item from the HSOPSC Version 2 was mistakenly used in place of the intended Version 1 item. Because the administered item did not reflect the original construct, it was excluded from the analysis to maintain conceptual validity. No sensitivity analysis was conducted, as the substituted item was conceptually unrelated to the intended construct. The score for this dimension was thus calculated based on the remaining three items.

Data entry was performed using Microsoft Excel. Descriptive statistics, including frequencies, percentages, and means, were used to summarize participants’ characteristics and survey responses. The differences in the dimensions’ PRRs were compared between the groups (e.g., PHs vs. DHs) using chi-square or Fisher’s exact tests, as the dimensional PRR was ordinal and limited to four or five discrete levels. The choice of statistical test was determined based on the distribution of the contingency tables. The chi-square test was applied when all expected cell frequencies were above five, whereas Fisher’s exact test was used in cases of small sample sizes or sparse data. The internal consistency of each dimension was assessed using Cronbach’s alpha (α) to evaluate the reliability of the scales. Previous studies using the HSOPSC in diverse cultural contexts have reported challenges in achieving high internal consistency, particularly in dimensions such as “Staffing” and “Nonpunitive Response to Error” [21,22]. Therefore, a threshold of α ≥ 0.60 was adopted as the minimum desirable level of reliability. Dimensions with α values below this threshold were retained in the analysis, but their results were interpreted with caution.

All statistical analyses were performed using Stata/BE version 18.

### 2.6. Ethical Considerations

Ethical approval was obtained from the relevant Japanese body (NCGM-S-004751-00, approved on 18 October 2023). The participants were given written information about the study and were asked to provide written informed consent. Participation in the study was voluntary. We used the HSOPSC developed by the AHRQ, which is publicly available for non-commercial use and does not require permission.

## 3. Results

Of the 335 questionnaires distributed, 289 were returned, resulting in an initial response rate of 86.3%. After excluding questionnaires that lacked completed informed consent forms, were otherwise incomplete, or had ineligible responses, 253 valid responses remained, yielding a final response rate of 75.5% (69.5% for PHs, 78.3% for DHs), as shown in Figure 1. The final response rates from each hospital ranged from 52% to 84% in PHs (average: 70%) and from 40% to 100% in the 23 DHs (average: 78%).

Table 2 shows the characteristics of the participants. Most participants were from DHs (70.6%), between the ages of 30 and 39 (47.1%), and female (53.4%). The participants were mainly physicians (39.4%), followed by nurses (27.7%), pharmacists (8.7%), midwives (7.5%), and dentists (2.8%). Approximately 30% of the participants had 6–10 years of experience at their current hospitals, while 19.8% had 1–5 years of experience. Less than one-third of the participants (29.7%) had experience in patient safety training. Due to variability in how participants interpreted the term “patient safety training,” further analysis based on training experience was not conducted.

Exclusion rates per dimension ranged from 2.4% and 8.3%, except for “Communication Openness,” which had a higher exclusion rate of 16.2%. Table 3 shows Cronbach’s α for the 12 dimensions of the HSOPSC, which ranged from 0.33 (“Management Support for Patient Safety”) to 0.7 (“Handoffs and Transitions” and “Frequency of Events Reported”).

Figure 2 presents the mean PRRs of each dimension. “Organizational Learning–Continuous Improvement” had the highest mean PRR (79.8%), while “Nonpunitive Response to Error” had the lowest (18.0%).

Figure 3 compares the mean PRRs of each item of the dimensions between PHs and DHs by the chi-square or Fisher’s exact tests. It reveals that PHs generally scored higher than DHs. PHs scored significantly higher in “Supervisor/Manager Expectations and Actions Promoting Patient Safety” (73.1% vs. 61.2%, *p* = 0.01) and “Handoffs and Transitions” (44.6% vs. 38.1%, *p* = 0.02). Conversely, DHs scored significantly higher in “Overall Perceptions of Patient Safety” (35.2% vs. 27.3%, *p* < 0.01).

Table 4 compares the mean PRRs of the different dimensions between the major professions represented in the sample: physicians, nurses, pharmacists, and midwives. “Nonpunitive Response to Error” and “Frequency of Events Reported” received the lowest percentage of positive responses across all professions. Each profession showed weaknesses in at least one dimension. For example, physicians revealed the lowest percentage for “Organizational Learning–Continuous Improvement” (76.4%), and nurses revealed the lowest percentage for “Overall Perceptions of Patient Safety” (30.4%). Meanwhile, midwives showed the lowest percentage for “Feedback and Communication About Error” (35.1%), and pharmacists showed the lowest percentage for “Nonpunitive Response to Error” (9.1%). Across the dimensions, “Management Support for Patient Safety” demonstrated a statistically significant difference among the professions (*p* = 0.04). Pharmacists responded the most positively to “Management Support for Patient Safety” (65.1%), whereas physicians, nurses, and midwives reported percentages in the 40s.

## 4. Discussion

This is the first study exploring PSC in Laos, and it highlights the strengths and challenges across PHs and DHs. According to the HSOPSC User’s Guide, a dimension with a mean PRR of 75% or higher is considered an organizational strength, whereas those below 50% indicate areas needing improvement [2]. Strengths were identified in dimensions like “Organizational Learning–Continuous Improvement” (79.8%). This may be partly explained by ongoing quality improvement support in the four southern provinces through JICA’s technical cooperation, along with national efforts to institutionalize hospital-based quality committees and action planning as part of accreditation preparation. However, barriers were revealed to creating an environment that supports open error reporting and fosters nonpunitive responses, as evidenced by the extremely low scores in the “Nonpunitive Response to Error” (18.0%) and “Frequency of Events Reported” (18.4%) dimensions. These findings underscore the need for targeted interventions to address these gaps, particularly through initiatives that cultivate a nonpunitive culture and strengthen error reporting systems to enhance transparency and learning from errors.

Healthcare workers’ fear of reporting errors can lead to under-reporting, resulting in missed opportunities to recognize and improve breakdowns and mitigate the risk of error recurrence [23,24,25,26]. A culture of blame has been the main weakness indicated in previous studies [27]. Table 5 compares several studies that used the HSOPSC and reveals that “Nonpunitive Response to Error” and “Error Reporting” tend to be rated lower [14,28,29,30,31,32,33]. The reason underlying healthcare workers’ reluctance to report errors is usually a fear of punishment [25]. In environments with a strong culture of blame, healthcare workers are more likely to avoid reporting errors, which ultimately weakens the PSC. Studies have found that blame culture is a major obstacle to PSC, and reducing errors requires systemic interventions [6].

Establishing a nonpunitive error reporting culture requires essential leadership support. Leaders who encourage error reporting and create an environment that protects those who report can provide a foundation for increasing error reporting and thus build a stronger PSC [7]. When strong leadership support is provided, healthcare workers are more likely to report errors without fear, fostering a culture of learning from mistakes.

A comparison of the percentages of positive responses between the PHs and DHs revealed notable differences in several aspects of their PSCs. Contrary to previous studies in other countries, which have reported that higher-level hospitals tend to show lower levels of perceived PSC due to factors such as increased patient complexity and workloads [34,35], the findings from Laos demonstrated that PHs generally scored higher than DHs across most dimensions. This discrepancy could be explained by key contextual differences in healthcare system structure and resource allocation between countries. Laos has a lower population density than many Southeast Asian countries, and the number of patients at the PHs in this study was relatively modest [18] compared with those in more urbanized countries. Moreover, DHs in rural Laos frequently experience chronic shortages of essential medicines and qualified healthcare workers [15,16], leading to operational difficulties and limited capacity to implement or sustain patient safety initiatives. The study was conducted during a period of significant economic instability in Laos, characterized by major inflation and budget issues [36]. These factors led to widespread medicine and workforce shortages, especially in DHs [37]. These systemic challenges could also create an environment in which DH staff perceive lower levels of PSC, even in the absence of complex tertiary-level care responsibilities.

In this study, PHs significantly outperformed DHs in “Supervisor/Manager Expectations and Actions Promoting Patient Safety” (73.1% vs. 61.2%, *p* = 0.01) and “Handoffs and Transitions” (44.6% vs. 38.1%, *p* = 0.02). These results suggest that PHs likely benefit from stronger managerial oversight and more structured patient care transition processes. The approximately 12% higher score for “Supervisor/Manager Expectations and Actions Promoting Patient Safety” in PHs compared with DHs (73.1% vs. 61.2%) implies that PHs may benefit from more formalized leadership structures and clearer supervisory roles. In contrast, DHs may lack consistent managerial engagement or structured protocols, which can affect staff’s confidence in leadership support for safety initiatives. The comparison with prior HSOPSC-based studies in countries such as the United States, Vietnam, and Japan indicates that the PHs in our study showed relatively higher mean PRRs for this dimension, while DHs reported lower scores. These findings indicate that managerial support for patient safety may require strengthening in DHs (Table 5).

Conversely, DHs scored significantly higher than PHs in “Overall Perceptions of Patient Safety” (35.2% vs. 27.3%, *p* < 0.01). This finding could indicate that despite resource constraints and operational challenges, DH staff perceived their work environment as safer, possibly due to smaller team sizes and closer interpersonal relationships, which can foster a sense of shared responsibility and trust. These disparities illustrated the need for tailored interventions that address the unique challenges faced by each hospital type. In PHs, efforts could focus on enhancing frontline staff engagement to complement existing managerial strengths, while in DHs, strategies may include strengthening organizational support and resources to ensure that positive perceptions translate into measurable improvements in patient safety practices.

A comparison of mean PRRs among professions in Ethiopia revealed that “Management Support for Patient Safety” demonstrated statistically significant differences among professions (*p* = 0.04), with pharmacists reporting the highest positive response rate (65.1%); notably, nurses’ perception of PSC was higher compared with that of other professions [33]. This difference could be attributed to each profession’s distinct roles, responsibilities, and work environments between countries. For instance, in provincial and DHs in Laos, pharmacists often play well-defined roles in medication management and may receive more direct support from the management. Conversely, physicians and nurses, who are responsible for broader patient care and often face the direct consequences of their errors, may perceive a lack of managerial support. These findings indicated the importance of addressing specific professional needs to effectively improve PSC. However, the internal consistency of the “Management Support for Patient Safety” dimension in our study was relatively low (Cronbach’s alpha = 0.33). Therefore, these results should be interpreted with caution, and further studies should seek to improve the measurement reliability to better assess perceptions of managerial support. Across all professions, “Nonpunitive Response to Error” and “Frequency of Events Reported” scored the lowest, indicating that concerns about error reporting were a common challenge.

Team-based training programs (e.g., Comprehensive Unit-Based Safety Program, Team Strategies and Tools to Enhance Performance and Patient Safety, and Crew Resource Management), incident reporting systems, feedback mechanisms, management engagement, leadership development, and involvement of patients and frontline staff have all been shown to strengthen PSC, and multiple interventions have been recommended [6,7,23]. Just Culture initiatives were among the interventions that aimed to improve staff perceptions of nonpunitive responses to error, and this training, which emphasizes fair accountability and system-based learning from errors, is effective in improving “Nonpunitive Response to Error” [7,23]. In this study, both “Nonpunitive Response to Error” and “Frequency of Events Reported” scored low, and to address the challenges identified in this study, the following priorities may be relevant for the four southern provinces of Laos: First, hospitals should establish a clear and structured incident reporting system that protects confidentiality and encourages transparency. Second, implementing Just Culture training for nurse managers, who often play a key role in daily operations, can build leadership capacity to support a nonpunitive environment.

In line with findings from other studies, this study shows that implementing standardized tools like the HSOPSC effectively highlights strengths and areas needing improvement. While this study showed a low Cronbach’s α for the 12 dimensions, numerous studies have shown low internal consistencies, especially in the dimensions of “Staffing,” “Communication Openness,” “Nonpunitive Response to Error,” “Organizational Learning,” and “Overall Perceptions of Safety” [11]. Although the HSOPSC is widely used worldwide, its application requires careful consideration of each country’s unique healthcare environment and cultural influences [11,28].

### Strengths and Limitations

A major strength of this study is its pioneering nature, as it represents the first comprehensive assessment of PSC in Laos, providing critical baseline data to inform future improvements. Plans are in place to conduct periodic reassessments using the HSOPSC to monitor changes over time and support evidence-based policy development. Additionally, using the HSOPSC could allow for future comparisons with international findings, offering valuable insights into areas where hospitals in Laos either aligned with or diverged from global standards. Although this study was limited to public PHs and DHs in Laos’s four southern provinces, which had already established patient safety committees and may reflect a more developed safety culture, the findings may still be applicable to other regions in Laos. This is because the national healthcare system is uniformly structured, and the Ministry of Health is currently promoting patient safety nationwide by establishing new quality standards that require all hospitals to implement incident reporting systems and safety-related initiatives. As these efforts expand, experiences from the four provinces may serve as a model for other areas.

However, certain limitations should be acknowledged. First, because the participants in this study were members of patient safety committees, their responses may not have fully represented the perspectives of the broader hospital staff. Patient safety committee members may be more aware of or involved in patient safety issues than other staff, which could influence their perceptions. Potential selection bias cannot be ruled out, as we could not compare characteristics of respondents and non-respondents due to the anonymous nature of the survey. Although the questionnaires were distributed to patient safety committee members, we could not identify the individuals who responded. Second, the questionnaires were distributed and collected through hospital administrators, which may have allowed for social desirability bias to occur. Participants may have been inclined to provide more favorable responses. To minimize this possibility, the survey was conducted anonymously, and participants were informed that their individual responses would be kept confidential and used only for research purposes. Third, the nature of the study may have limited the generalizability of the findings to the parts of Laos with central hospitals or other regions of the country with different healthcare contexts or resource availability. Fourth, the study was conducted during a period of significant economic instability, characterized by a sharp depreciation of the Lao Kip, rising inflation, and national budget constraints. Future studies would benefit from different timing and from including a wider range of staff roles and hospital types, such as central hospitals in Vientiane. Furthermore, this study applied only three items for the “Teamwork Within Units” dimension instead of the standard four, and this may have impacted the comparability of the results with other studies using all four items. Another limitation concerns the internal consistency of several dimensions, with Cronbach’s alpha values falling below the commonly accepted threshold of 0.60. In countries where the concept of PSC is still emerging, such as Laos, respondents may interpret certain items inconsistently. Although we conducted comprehension checks and feedback-based reviews to adapt the questionnaire to the context of Laos, formal pretesting with a sufficient sample size was not conducted due to time constraints. Consequently, shared understandings of key terms and constructs may not have been achieved across hospitals with varying capacities and resources, potentially contributing to low internal consistency. Additionally, while we used bivariate tests to explore group differences, the absence of regression-based adjustment limits the ability to draw inferences about independent associations between variables. Regression-based adjustments could not be made owing to the relatively small sample size and the exploratory nature of the study. Future studies with re-validated questionnaires and larger samples should consider using multivariate models to control for potential confounders and enhance the validity of intergroup comparisons.

## 5. Conclusions

This study was the first to investigate PSC in southern Laos using the HSOPSC tool. It illuminated local strengths in organizational learning while identifying nonpunitive responses and event reporting as critical areas for improvement. To address these challenges, hospitals should foster an environment in which staff feel safe to report errors without fear of blame. Strengthening leadership support and reporting systems is essential. Practical strategies could include Just Culture training for head nurses and the development of clear and confidential incident reporting channels. While this study provides valuable insight, its findings should be interpreted with caution due to several limitations: internal consistency was low in some dimensions, the sample consisted solely of committee members, and analyses were based on bivariate methods without adjustment for confounders. Nevertheless, these exploratory findings offer practical entry points for future interventions in Lao hospitals and other low-resource settings. Further research is needed using a refined Lao version of the questionnaire, involving larger and more diverse samples, as well as qualitative and intervention-based approaches.

## Figures and Tables

**Figure 1 healthcare-13-01934-f001:**
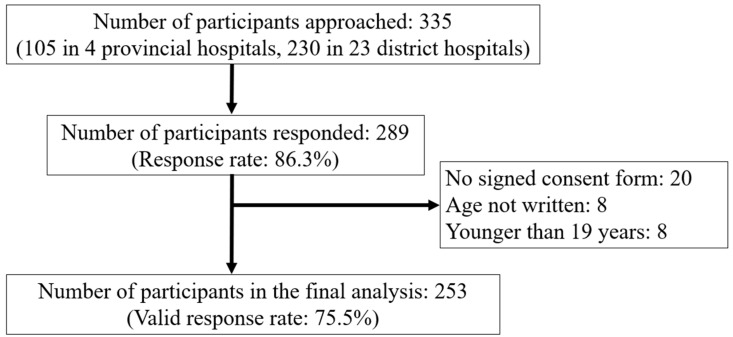
Flow diagram of study participants.

**Figure 2 healthcare-13-01934-f002:**
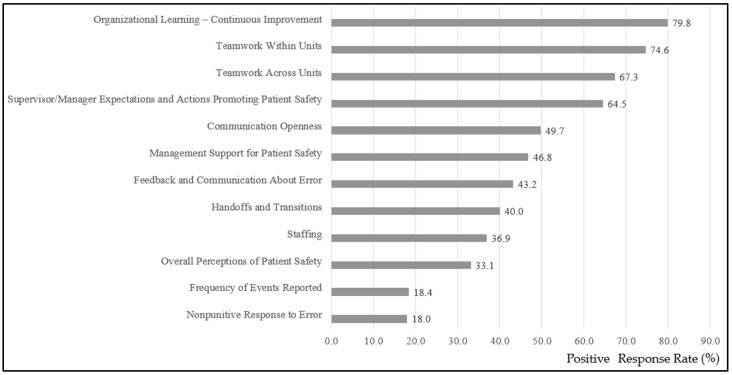
Mean positive response rates for each item across the 12 dimensions in all 27 hospitals.

**Figure 3 healthcare-13-01934-f003:**
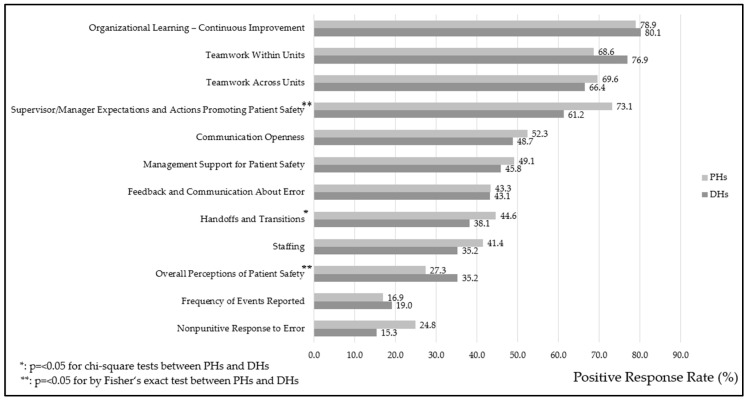
Mean positive response rates for each item across the 12 dimensions in public provincial hospitals (PHs) and district hospitals (DHs).

**Table 1 healthcare-13-01934-t001:** Dimensions and items of the HSOPSC.

Dimension	Number of Items	Example of Item in HSOPSC
Supervisor/Manager Expectations and Actions Promoting Patient Safety	4	My supervisor/manager seriously considers staff suggestions for improving patient safety.
Organizational Learning–Continuous Improvement	3	Mistakes have led to positive changes here.
Teamwork Within Units	4 *	People support one another in this unit.
Communication Openness	3	Staff will freely speak up if they see something that may negatively affect patient care.
Feedback and Communication About Error	3	We are given feedback about changes put into place based on event reports
Nonpunitive Response to Error	3	Staff feel like their mistakes are held against them. (negatively worded)
Staffing	4	We use more agency/temporary staff than is best for patient care. (negatively worded)
Management Support for Patient Safety	3	Hospital management provides a work climate that promotes patient safety.
Teamwork Across Units	4	There is good cooperation among hospital units that need to work together.
Handoffs and Transitions	4	Shift changes are problematic for patients in this hospital. (negatively worded)
Overall Perceptions of Patient Safety	4	We have a patient safety problem in this unit. (negatively worded)
Frequency of Events Reported	3	When a mistake is made that could harm the patient, but does not, how often is this reported?

* The dimension “Teamwork Within Units” was ultimately evaluated using three items in this study, although the HSOPSC contains four items.

**Table 2 healthcare-13-01934-t002:** Participant characteristics.

Characteristics	Number (*n* = 253)	Percentage
Age		
20–29 years	47	18.6%
30–39 years	119	47.0%
40–49 years	52	20.6%
50–59 years	32	12.7%
60 years or older	3	1.2%
Gender		
Men	69	27.3%
Women	135	53.4%
Unknown/Missing	49	19.4%
Profession		
Physician	100	39.5%
Nurse	70	27.7%
Pharmacist	22	8.7%
Midwife	19	7.5%
Dentist	7	2.8%
Others	32	12.7%
Unknown/Missing	3	1.2%
Patient Safety Training Experience		
Yes	75	29.6%
No	169	66.8%
Unknown/Missing	9	3.6%
Years of Experience at Current Hospital		
Less than 1 year	9	3.6%
1–5 years	50	19.8%
6–10 years	79	31.2%
11–15 years	39	15.4%
16–20 years	28	9.9%
21 years or more	43	17.0%
Unknown/Missing	5	2.0%
Type of Current Hospital		
Provincial hospital	73	29.4%
District hospital	180	70.6%

**Table 3 healthcare-13-01934-t003:** Cronbach’s α for the 12 dimensions of the HSOPSC.

Dimension	Number of Items	Cronbach’s α
Supervisor/Manager Expectations and Actions Promoting Patient Safety	4	0.51
Organizational Learning–Continuous Improvement	3	0.64
Teamwork Within Units *^1^	3	0.43
Communication Openness	3	0.41
Feedback and Communication About Error	3	0.65
Nonpunitive Response to Error	3	0.54
Staffing	4	0.37
Management Support for Patient Safety	3	0.33
Teamwork Across Units	4	0.54
Handoffs and Transitions	4	0.70
Overall Perceptions of Patient Safety	4	0.39
Frequency of Events Reported	3	0.68

*^1^ This dimension had four items in the HSOPSC.

**Table 4 healthcare-13-01934-t004:** Comparison of the mean positive response rates across dimensions by profession.

Dimension	Physicians (*n* = 100) (%)	Nurses (*n* = 70) (%)	Pharmacists (*n* = 22) (%)	Midwifes (*n* = 19) (%)	*p*-Value
Nonpunitive Response to Error	16.0	18.0	9.1	18.5	0.96
Frequency of Events Reported	19.4	15.4	22.7	11.8	0.58
Overall Perceptions of Patient Safety	35.0	30.4	36.9	30.9	0.53
Staffing	34.3	38.7	39.8	38.9	0.90
Handoffs and Transitions	38.0	44.1	39.3	34.7	0.74
Feedback and Communication About Error	45.9	42.1	58.3	35.1	0.62
Management Support for Patient Safety	43.6	47.3	65.1	44.4	0.04 *
Communication Openness	48.9	56.4	53.7	44.4	0.60
Supervisor/Manager Expectations and Actions Promoting Patient Safety	64.0	67.8	61.9	55.9	0.24
Teamwork Across Units	64.6	70.0	72.7	66.2	0.65
Teamwork Within Units	76.0	74.1	77.3	75.4	0.78
Organizational Learning–Continuous Improvement	76.4	86.9	86.4	83.3	0.18

* A significant difference was noted between the four professions using Fisher’s exact test (*p* = 0.0035).

**Table 5 healthcare-13-01934-t005:** Comparison of 12 dimensions’ mean positive response rates between studies.

Dimension	This Study	Taiwan (2010) [28]	Lebanon (2010) [32]	Japan (2013) [30]	Ethiopia (2017) [33]	USA (2018) [29]	China (2020) [31]	Vietnam (2021) [14]
Nonpunitive Response to Error	18%	45%	24%	43%	33%	47%	36%	53%
Frequency of Events Reported	18%	57%	68%	66%	36%	67%	72%	77%
Overall Perceptions of Patient Safety	33%	65%	73%	53%	26%	66%	64%	75%
Staffing	37%	39%	37%	40%	26%	53%	38%	49%
Handoffs and Transitions	40%	48%	50%	36%	33%	48%	57%	63%
Feedback and Communication About Error	43%	59%	68%	53%	46%	69%	87%	83%
Management Support for Patient Safety	47%	62%	78%	51%	47%	72%	36%	85%
Communication Openness	50%	58%	57%	49%	42%	66%	64%	66%
Supervisor/Manager Expectations and Actions Promoting Patient Safety	64%	83%	66%	61%	46%	80%	78%	86%
Teamwork Across Units	67%	72%	56%	44%	57%	62%	64%	73%
Teamwork Within Units	75% *	94%	56%	70%	72%	82%	87%	91%
Organizational Learning–Continuous Improvement	80%	84%	78%	51%	72%	72%	87%	88%

* This dimension was evaluated using three items in this study.

## Data Availability

The data presented in this study are not publicly available due to ethical and privacy restrictions, as they contain sensitive information regarding individual participants and affiliated hospitals. Access to the data may be considered upon reasonable request and with appropriate ethical approval.

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
