# Peer review of "Patient Safety Culture of Hospitals in Southern Laos: A Cross-Sectional Study Using the Hospital Survey on Patient Safety Culture"

_healthcare, 2025, doi:10.3390/healthcare13151934_

Round 1
Reviewer 1 Report
Comments and Suggestions for Authors
Dear authors,
Thank you for the opportunity to review the manuscript.
Beginning with the abstract, authors can separate the background from the objectives, making it clearer what is already known about the topic (e.g., what previous studies indicate about patient safety culture in similar contexts) and the specific relevance of this study in Laos. The objective of the study should be stated clearly and concisely to reinforce the contribution of the work.
It is suggested to avoid the use of acronyms, such as PSC (Patient Safety Culture), in the abstract. Repeated use of acronyms can make the abstract difficult to read and understand, especially for readers who are unfamiliar with the topic.
It is also recommended to reorganise the Methods, Results and Conclusions sections to ensure greater clarity and objectivity, shortening overly long sentences and avoiding excessively detailed information that could be left for the body of the article (e.g. multiple percentage values and statistical tests). In the abstract, the focus should be on the most relevant findings and their interpretation.
In the Conclusion, it is recommended to reinforce the practical implications of the findings and, if possible, to point out a guideline for clinical practice or future research.
Overall, the abstract could benefit from clearer writing, with better structuring of ideas and less density of numbers, facilitating quick reading and comprehension by the reader.
The keywords chosen are relevant to the study but do not correspond to the indexed terms or Mesh. The choice of key terms in natural language can make it difficult to present your article when searched in the database using indexed terms, reducing its recognition and dissemination.
The introduction presents a logical sequence on the relevance of patient safety culture and its relationship with the quality and safety of care. However, it could benefit from greater segmentation of the paragraphs to make it easier to read.
For example: An initial paragraph focused on the definition and overall importance of PSC. A second paragraph on existing assessment tools (e.g., HSOPSC). A third paragraph focused on the specific context of Southeast Asia and Laos, highlighting the evidence gap.
The authors present the general and regional context well, but it would be useful to emphasise more clearly the specific gap in the literature in Laos, perhaps with data on local safety indicators, if available. Better differentiate between what is known from studies in neighbouring countries (e.g. Vietnam) and what is unknown in Laos, to reinforce the relevance and novelty of the study.
Although the objective is mentioned at the end, it is recommended to highlight it in a more direct and isolated sentence that the reader can immediately identify. For example: “This study aimed to evaluate the state of patient safety culture in hospitals in southern Laos using the HSOPSC.”
I also recommend avoiding overly long sentences, which can make it difficult to follow the main ideas. Review minor redundancies in the text (e.g., when repeating the need for research in Laos at different points). Consider briefly mentioning the potential implications of the study in the introduction to create a link to practical relevance.
The methods present relevant information, but greater objectivity and focus on the study design are recommended:
🔹 Separate the national context from the study design.
Example: the opening paragraph (“Laos is a lower-middle-income country...”) contains extensive details about the health system (administrative levels, role of the Ministry of Health), which could be summarised or integrated into the introduction. This would make the section more focused on the methodological aspects themselves.
🔹 Clarify and highlight essential elements of the study.
Example: the sentence “A hospital-based cross-sectional study was conducted in the four provinces...” could be the beginning of the methods section, as it contains the central information. It is suggested to bring it to the beginning and then detail the necessary context.
🔹 Eliminate or condense excessive details.
Example: ‘On average, one PH has 131 beds with 199 permanent healthcare workers, and one DH has 20 beds with 28 permanent healthcare workers.’ these figures can be summarised or presented only if they are relevant to the data analysis.
🔹 Clarify and simplify the sampling method
The text states: ‘The study used a purposive sampling method to recruit healthcare workers who served on each hospital's patient safety committee.’ This key point could be highlighted more directly and concisely at the beginning of the paragraph, avoiding repetition of the inclusion and exclusion criteria.
Example: The sentence “Healthcare workers who were members of such committees at the target hospitals and who provided informed consent were eligible participants” could be integrated into the opening sentence for a more concise presentation.
🔹 Reduce excessive detail on sample planning
The final paragraph provides many numbers and details about the distribution of participants: “30 participants were recruited from the Champasak PH, 25 from each of the other three PHs (Attapeu, Sekong, Salavan), and 10 from each of the 23 DHs…” This level of detail could be summarised in the text and presented in a table or note to improve the flow of the text.
🔹 Avoid repetition
The idea that professionals were recruited because they had roles related to patient safety appears more than once (e.g., “healthcare workers who played diverse roles directly related to patient safety was deemed appropriate…”). We suggest condensing this information into a single sentence.
🔹 Insufficient description of cultural and linguistic validation
Although the authors mention translation, back-translation and pre-testing, there is a lack of detail on: How the semantic and conceptual equivalence of the items was assessed (beyond clarity and comprehension).
Whether validity indicators (e.g., content validity by expert panel) were calculated during the adaptation process.
How changes made to the questionnaire were handled (e.g., “reworded questions for clarity and added brief explanatory notes”). The impact of these changes on the comparability of results with other studies using the HSOPSC is not discussed.
🔹 Modification of the scale is not justified
The note in Table 1 indicates that the dimension ‘Teamwork within Units’ was assessed with three items (instead of four). The reason for this modification and its impact on the validity and reliability of the dimension are not explained. The reviewer expects to see justification when a validated instrument is changed.
🔹 Lack of information on reliability in the target sample
No data are presented on the internal consistency (e.g., Cronbach's alpha) of the questionnaire or its dimensions in the sample studied. This is a relevant limitation because cultural adaptation can impact reliability.
🔹 Pre-test lacking in detail
The pre-test is mentioned, but: It is not known how many participants took part in it.
It is not clear whether the pre-test was only qualitative (e.g. cognitive interviews) or whether quantitative data was collected.
Feedback on Data Collection:
🔹 Potential risk of bias in data collection: The method described (questionnaires delivered via various hierarchies — directors, committee heads, public health authorities) may introduce:
Response bias due to institutional pressure: participants may feel compelled to respond in a certain way, knowing that the questionnaires pass through several hands before reaching the research team (despite being sealed).
Risk of perceived breach of anonymity: it is not entirely clear how participants were assured that their responses would remain confidential.
The authors are advised to comment on this potential bias and, if applicable, how it was mitigated (e.g., explicit communication about anonymity, how the process was explained to participants).
Feedback on Analysis:
🔹 Elimination of item in dimension calculation – lack of detail on impact
The authors state: “one item in the ‘Teamwork within Units’ dimension was found to differ in meaning [...] and was therefore excluded from the analysis.”
It would be important to explain how the exclusion of this item impacted comparability with other studies using the HSOPSC.
Detail the results of the sensitivity analysis referred to (it is not enough to say that it was performed — the reader expects to know what the test indicated).
🔹 Criteria for dealing with missing data
The text states: “participants with missing responses for any items within a dimension were excluded from these calculations.”
This criterion is very restrictive and can lead to significant data loss. The authors should justify why they did not use alternative approaches (e.g., imputation, average of responses given) to avoid bias due to case exclusion.
Indicate the proportion of data/exclusions per dimension due to missing data.
🔹 Statistical analysis — lack of detail on analytical model
Although the tests used are mentioned (chi-square/Fisher), it is necessary to clarify:
Whether there was adjustment for potential confounding factors (e.g., type of hospital may be associated with other variables such as size, resources).
Why only bivariate tests were chosen and not multivariate analyses, given the objective of the study.
🔹 Reliability assessment: The authors used Cronbach's alpha with a threshold of 0.60. However it is necessary to present the values obtained by dimension (not just mention the threshold).
It would be relevant to discuss the impact of low alphas on the interpretation of the results, in addition to stating that they were ‘interpreted with caution’.
The results section:
🔹 The authors compare dimensions between types of hospitals and professions using bivariate tests (chi-square, Fisher). It would be methodologically more robust to apply multivariate analyses (e.g., logistic regression or generalised linear models) to adjust for potential confounding factors (e.g., age, gender, type of hospital, years of experience). The absence of this adjustment limits the causal interpretation and generalisation of the findings.
🔹 The alpha values presented range from 0.33 to 0.7. Although the authors mention interpreting dimensions with alpha < 0.60 with caution, they fail to discuss the concrete impact of these low values on the reliability of the results presented (e.g., Management Support for Patient Safety with 0.33). This could affect the validity of the findings reported in these dimensions.
🔹 The participant flow shows that 253 (75.5%) were included in the final analysis. It would be relevant to report whether there were systematic differences between those included and those excluded (e.g., by type of hospital, profession) to assess the risk of selection bias.
🔹 The results are mainly descriptive (percentages by dimension and groups), but, a more in-depth analysis of the magnitude and practical relevance of the observed differences is lacking (e.g., what does a difference of ~12% in Supervisor/Manager Expectations between PHs and DHs mean in practice?). Furthermore, there is no discussion of whether these differences are consistent with previous studies or with HSOPSC benchmarks.
🔹 It is reported that less than a third of participants had patient safety training.
The relationship between this variable and perceptions of safety culture could be explored, given its potential explanatory role.
In the discussion the authors compare types of hospitals and professions, but the discussion does not address the relevant limitation: the differences observed may be associated with uncontrolled factors (e.g., age, experience, previous training). It is suggested to discuss that the absence of multivariate analyses limits the strength of the conclusions and that future studies should adjust for confounding variables.
Although the discussion refers to low Cronbach's alpha values (e.g., Management Support for Patient Safety 0.33), there is no critical analysis of what this implies for the interpretation of data from these dimensions. It is important to recognise that the results of these dimensions should be read with caution, as low internal consistency may compromise the validity of the conclusions.
The authors state that the results may apply to other regions of Laos. However, as they acknowledge in the limitations section, the study only included hospitals in the south and safety committee professionals.
The impact of excluding an item from the Teamwork within Units dimension is minimised, as it is said not to affect the central conclusions. It would be more rigorous to discuss how this exclusion may limit comparability with international studies and affect the measurement of the dimension.
Although leadership and a non-punitive culture are identified as necessary, more operational suggestions are lacking (e.g., specific training, strengthening adverse event reporting systems, benchmarking initiatives).
The conclusion is underdeveloped; further development of this section is recommended.
The final paragraph does not sufficiently reflect the methodological limitations identified (e.g., low reliability of some dimensions, sample restricted to committee members, lack of adjustment for confounders). This could lead to an overestimation of the conclusions. The conclusions should be moderate and acknowledge the limitations that condition the generalisation of the findings.
The text mentions strengthening leadership to foster a non-punitive culture, but does not translate these recommendations into concrete practical actions (e.g., training, review of incident reporting policies, creation of safe reporting channels). The authors could make the recommendations more applicable and specific to the context studied.
When stating that the results can improve the quality of healthcare, there is a lack of direct evidence in the study to support this association (quality indicators were not measured). They can be reformulated to reflect that these are hypotheses that should be tested in future interventions.
In conclusion, this study provides valuable baseline data on patient safety culture in Laos and highlights important strengths and areas for improvement, notably in teamwork and organisational learning, as well as in non-punitive responses and error reporting. However, several methodological limitations, including low internal consistency of some dimensions, the restricted sample (limited to members of the patient safety committee), the absence of multivariate analyses, and potential biases in data collection, limit the strength and generalisability of the results. The discussion and conclusions could benefit from more critical reflection on these issues and a clearer link between the results and practical recommendations.
Based on the above assessment, I recommend a major revision, as the manuscript presents interesting and relevant data but requires substantial improvements in methodological transparency, interpretation of results, and critical discussion to meet publication standards.
Author Response
3. Point-by-point response to Comments and Suggestions for Authors |
||||||||||||||||||||||||||||||||||||||||||||||||||||||||||||||||||||||||||||||||||||||||||||||||||||||||||||||||||||||||||||||||||||||||||||
Comments 1: Beginning with the abstract, authors can separate the background from the objectives, making it clearer what is already known about the topic (e.g., what previous studies indicate about patient safety culture in similar contexts) and the specific relevance of this study in Laos. The objective of the study should be stated clearly and concisely to reinforce the contribution of the work. |
||||||||||||||||||||||||||||||||||||||||||||||||||||||||||||||||||||||||||||||||||||||||||||||||||||||||||||||||||||||||||||||||||||||||||||
Response 1: Thank you for your observation. We agree with your comment and have separated the background from the objective in the Abstract to clarify what is already known about patient safety culture in similar contexts and to outline the relevance of this study in Laos. The revised objective is now clearly stated in the Abstract (Lines 12–17 of the revised manuscript). |
||||||||||||||||||||||||||||||||||||||||||||||||||||||||||||||||||||||||||||||||||||||||||||||||||||||||||||||||||||||||||||||||||||||||||||
Comments 2: It is suggested to avoid the use of acronyms, such as PSC (Patient Safety Culture), in the abstract. Repeated use of acronyms can make the abstract difficult to read and understand, especially for readers who are unfamiliar with the topic. |
||||||||||||||||||||||||||||||||||||||||||||||||||||||||||||||||||||||||||||||||||||||||||||||||||||||||||||||||||||||||||||||||||||||||||||
Response 2: We agree with your suggestion and have revised the Abstract to replace the acronym “PSC” with the full term “patient safety culture” throughout (Lines 12–17). |
||||||||||||||||||||||||||||||||||||||||||||||||||||||||||||||||||||||||||||||||||||||||||||||||||||||||||||||||||||||||||||||||||||||||||||
Comments 3: It is also recommended to reorganize the Methods, Results, and Conclusions sections to ensure greater clarity and objectivity, shortening overly long sentences and avoiding excessively detailed information that could be left for the body of the article (e.g., multiple percentage values and statistical tests). In the abstract, the focus should be on the most relevant findings and their interpretation. |
||||||||||||||||||||||||||||||||||||||||||||||||||||||||||||||||||||||||||||||||||||||||||||||||||||||||||||||||||||||||||||||||||||||||||||
Response 3: We appreciate the reviewer’s recommendation. We have revised the Methods and Results sections in the Abstract to improve clarity and brevity. Long sentences have been shortened, and excessive numerical data have been removed. Only the key findings are retained (Lines 17–32). |
||||||||||||||||||||||||||||||||||||||||||||||||||||||||||||||||||||||||||||||||||||||||||||||||||||||||||||||||||||||||||||||||||||||||||||
Comment 4: In the Conclusion, it is recommended to reinforce the practical implications of the findings and, if possible, to point out a guideline for clinical practice or future research. |
||||||||||||||||||||||||||||||||||||||||||||||||||||||||||||||||||||||||||||||||||||||||||||||||||||||||||||||||||||||||||||||||||||||||||||
Response 4: Thank you for your valuable suggestion. We have revised the Conclusion section of the Abstract to highlight the practical implications of our findings and added a recommendation for clinical practice (Lines 28–31). |
||||||||||||||||||||||||||||||||||||||||||||||||||||||||||||||||||||||||||||||||||||||||||||||||||||||||||||||||||||||||||||||||||||||||||||
Comment 5: Overall, the abstract could benefit from clearer writing, with better structuring of ideas and less density of numbers, facilitating quick reading and comprehension by the reader. |
||||||||||||||||||||||||||||||||||||||||||||||||||||||||||||||||||||||||||||||||||||||||||||||||||||||||||||||||||||||||||||||||||||||||||||
Response 5: Thank you for your insightful feedback on the Abstract's structure and clarity. We agree with your comments (1–5) and have made substantial revisions to improve flow, reduce complexity, and enhance readability. We have also restructured the Methods, Results, and Conclusion sections and reduced the volume of numerical data as advised. These updates are reflected in Responses 1–5. The revised Abstract is now clearer, more concise, and effectively presents the study’s key findings and implications. |
||||||||||||||||||||||||||||||||||||||||||||||||||||||||||||||||||||||||||||||||||||||||||||||||||||||||||||||||||||||||||||||||||||||||||||
Comment 6: The keywords chosen are relevant to the study but do not correspond to the indexed terms or Mesh. The choice of key terms in natural language can make it difficult to present your article when searched in the database using indexed terms, reducing its recognition and dissemination. |
||||||||||||||||||||||||||||||||||||||||||||||||||||||||||||||||||||||||||||||||||||||||||||||||||||||||||||||||||||||||||||||||||||||||||||
Response 6: Thank you for the recommendation. We have revised the keywords to align with standardized indexing terms such as MeSH, which should improve the article’s visibility and dissemination. The revised keywords are listed on Lines.33-34. Revised keywords: |
||||||||||||||||||||||||||||||||||||||||||||||||||||||||||||||||||||||||||||||||||||||||||||||||||||||||||||||||||||||||||||||||||||||||||||
Comment 7(Introduction): The introduction presents a logical sequence on the relevance of patient safety culture and its relationship with the quality and safety of care. However, it could benefit from greater segmentation of the paragraphs to make it easier to read. For example: An initial paragraph focused on the definition and overall importance of PSC. A second paragraph on existing assessment tools (e.g., HSOPSC). A third paragraph focused on the specific context of Southeast Asia and Laos, highlighting the evidence gap. The authors present the general and regional context well, but it would be useful to emphasize more clearly the specific gap in the literature in Laos, perhaps with data on local safety indicators, if available. Better differentiate between what is known from studies in neighboring countries (e.g. Vietnam) and what is unknown in Laos, to reinforce the relevance and novelty of the study. Although the objective is mentioned at the end, it is recommended to highlight it in a more direct and isolated sentence that the reader can immediately identify. For example: “This study aimed to evaluate the state of patient safety culture in hospitals in southern Laos using the HSOPSC.” I also recommend avoiding overly long sentences, which can make it difficult to follow the main ideas. Review minor redundancies in the text (e.g., when repeating the need for research in Laos at different points). Consider briefly mentioning the potential implications of the study in the introduction to create a link to practical relevance. |
||||||||||||||||||||||||||||||||||||||||||||||||||||||||||||||||||||||||||||||||||||||||||||||||||||||||||||||||||||||||||||||||||||||||||||
Response 7 (Introduction) :  Thank you for your detailed and constructive feedback on the Introduction. We appreciate the recommendation to restructure this section for improved clarity and flow. Accordingly, we have revised the Introduction by dividing it into distinct paragraphs as recommended.
We have also revised the final paragraph to clearly state the research objective in a standalone sentence. In addition, we have shortened long sentences, removed redundancies, and briefly outlined the potential implications of the study to enhance relevance. These changes appear on Lines 37–78 of the revised manuscript. |
||||||||||||||||||||||||||||||||||||||||||||||||||||||||||||||||||||||||||||||||||||||||||||||||||||||||||||||||||||||||||||||||||||||||||||
Comment 8 (Method): ? Separate the national context from the study design. Example: the opening paragraph (“Laos is a lower-middle-income country...”) contains extensive details about the health system (administrative levels, role of the Ministry of Health), which could be summarized or integrated into the introduction. This would make the section more focused on the methodological aspects themselves. ? Clarify and highlight essential elements of the study. Example: the sentence “A hospital-based cross-sectional study was conducted in the four provinces...” could be the beginning of the methods section, as it contains the central information. It is suggested to bring it to the beginning and then detail the necessary context. |
||||||||||||||||||||||||||||||||||||||||||||||||||||||||||||||||||||||||||||||||||||||||||||||||||||||||||||||||||||||||||||||||||||||||||||
Response 8 (Method): Thank you for your observation. We agree with the reviewer’s comment. Accordingly, we have removed the extended contextual description of Laos’s healthcare system from the Methods section and integrated a brief summary into the Introduction. The Methods section now begins with a direct description of the study design, improving clarity and objectivity. Essential elements are now clearly highlighted. These changes are on Lines 81–88. |
||||||||||||||||||||||||||||||||||||||||||||||||||||||||||||||||||||||||||||||||||||||||||||||||||||||||||||||||||||||||||||||||||||||||||||
Comment 9 (Method): ? Eliminate or condense excessive details. Example: ‘On average, one PH has 131 beds with 199 permanent healthcare workers, and one DH has 20 beds with 28 permanent healthcare workers.’ These figures can be summarized or presented only if they are relevant to the data analysis. ? Clarify and simplify the sampling method The text states: ‘The study used a purposive sampling method to recruit healthcare workers who served on each hospital's patient safety committee.’ This key point could be highlighted more directly and concisely at the beginning of the paragraph, avoiding repetition of the inclusion and exclusion criteria. Example: The sentence “Healthcare workers who were members of such committees at the target hospitals and who provided informed consent were eligible participants” could be integrated into the opening sentence for a more concise presentation. ? Reduce excessive detail on sample planning The final paragraph provides many numbers and details about the distribution of participants: “30 participants were recruited from the Champasak PH, 25 from each of the other three PHs (Attapeu, Sekong, Salavan), and 10 from each of the 23 DHs…” This level of detail could be summarized in the text and presented in a table or note to improve the flow of the text. ? Avoid repetition The idea that professionals were recruited because they had roles related to patient safety appears more than once (e.g., “healthcare workers who played diverse roles directly related to patient safety was deemed appropriate…”). We suggest condensing this information into a single sentence. |
||||||||||||||||||||||||||||||||||||||||||||||||||||||||||||||||||||||||||||||||||||||||||||||||||||||||||||||||||||||||||||||||||||||||||||
Response 9 (Method): Thank you for these detailed comments. In response, we have revised the Methods section to enhance clarity and conciseness as follows:
These revisions appear on Lines 81–99. |
||||||||||||||||||||||||||||||||||||||||||||||||||||||||||||||||||||||||||||||||||||||||||||||||||||||||||||||||||||||||||||||||||||||||||||
Comment 10 (Method: process explanation of cultural and linguistic validation): ? Insufficient description of cultural and linguistic validation Although the authors mention translation, back-translation, and pre-testing, there is a lack of detail on: How the semantic and conceptual equivalence of the items was assessed (beyond clarity and comprehension). Whether validity indicators (e.g., content validity by expert panel) were calculated during the adaptation process. How changes made to the questionnaire were handled (e.g., “reworded questions for clarity and added brief explanatory notes”). The impact of these changes on the comparability of results with other studies using the HSOPSC is not discussed. |
||||||||||||||||||||||||||||||||||||||||||||||||||||||||||||||||||||||||||||||||||||||||||||||||||||||||||||||||||||||||||||||||||||||||||||
Response 10 (Method: cultural and linguistic validation): Thank you for this insightful comment. We have added detailed descriptions of the translation and adaptation process in the revised Methods section (Lines 104–120) and Limitations (Lines 385–391). The English HSOPSC questionnaire was translated into Lao by a native speaker with medical translation experience. An independent native translator, blinded to the original, conducted a back-translation to assess semantic and conceptual equivalence. Discrepancies were resolved by the research team via discussion. A preliminary comprehension check was conducted with Lao-speaking physicians and nurses to assess clarity and contextual relevance. Additional feedback was obtained from physicians in the Department of Healthcare and Rehabilitation of the provincial health offices of the four participating provinces. Based on their input, brief explanatory notes were added. Notably, there is no exact Lao equivalent for the term “culture,” which is often interpreted as “history.” To address this, we included the HSOPSC definition in the translated questionnaire: “PSC refers to the shared values, beliefs, and norms about the importance of patient safety within an organization.” While formal validation metrics such as the Content Validity Index were not calculated due to the limited number of reviewers, feedback-based review and comprehension check by experienced healthcare workers were designed to preserve conceptual equivalence and cultural appropriateness. We have also acknowledged that such modifications may limit strict comparability with other HSOPSC studies and noted this as a limitation in the manuscript. |
||||||||||||||||||||||||||||||||||||||||||||||||||||||||||||||||||||||||||||||||||||||||||||||||||||||||||||||||||||||||||||||||||||||||||||
Comment 11 (Teamwork within Units): Method: ? Modification of the scale is not justified The note in Table 1 indicates that the dimension ‘Teamwork within Units’ was assessed with three items (instead of four). The reason for this modification and its impact on the validity and reliability of the dimension are not explained. The reviewer expects to see justification when a validated instrument is changed. Method: Feedback on Analysis: ? Elimination of item in dimension calculation – lack of detail on impact The authors state: “one item in the ‘Teamwork within Units’ dimension was found to differ in meaning [...] and was therefore excluded from the analysis.” It would be important to explain how the exclusion of this item impacted comparability with other studies using the HSOPSC. Detail the results of the sensitivity analysis referred to (it is not enough to say that it was performed — the reader expects to know what the test indicated).
In the Discussion session: The impact of excluding an item from the Teamwork within Units dimension is minimized, as it is said not to affect the central conclusions. It would be more rigorous to discuss how this exclusion may limit comparability with international studies and affect the measurement of the dimension. |
||||||||||||||||||||||||||||||||||||||||||||||||||||||||||||||||||||||||||||||||||||||||||||||||||||||||||||||||||||||||||||||||||||||||||||
Response 11 (Teamwork within unit): Thank you for these important observations. As stated in the original manuscript, “No sensitivity analysis was conducted, as the item was deemed conceptually incompatible with the construct; its inclusion would have compromised the validity of the dimension’s score.” This omission was due to a translation oversight rather than reliability metrics. Specifically, one item from HSOPSC Version 1 was inadvertently replaced with a semantically unrelated item from Version 2 during translation. Consequently, the item was excluded to preserve conceptual integrity. This clarification is now included in the revised Methods section (Lines 151–158). We acknowledge that this exclusion may reduce comparability with standard HSOPSC studies and have noted this as a limitation in the Discussion section. Moreover, based on HSOPSC guidelines, dimensions with ≥75% mean PRR are considered organizational strengths. Given this and the limitation noted above, we emphasized “Organizational Learning” over “Teamwork” in the Abstract and Conclusion. |
||||||||||||||||||||||||||||||||||||||||||||||||||||||||||||||||||||||||||||||||||||||||||||||||||||||||||||||||||||||||||||||||||||||||||||
Comment 12 (Method: process explanation of pre-test): ? Lack of information on reliability in the target sample No data are presented on the internal consistency (e.g., Cronbach's alpha) of the questionnaire or its dimensions in the sample studied. This is a relevant limitation because cultural adaptation can impact reliability. ? Pre-test lacking in detail The pre-test is mentioned, but it is not known how many participants took part in it. It is not clear whether the pre-test was only qualitative (e.g. cognitive interviews) or whether quantitative data were collected |
||||||||||||||||||||||||||||||||||||||||||||||||||||||||||||||||||||||||||||||||||||||||||||||||||||||||||||||||||||||||||||||||||||||||||||
Response 12 (Method): Thank you for your thoughtful comment. We agree that the term “pre-test” was inaccurate. We have replaced it with a more precise description, stating that a preliminary comprehension check was conducted with a small number of Lao-speaking physicians and nurses and feedback-based review was conducted by provincial health directors. Internal consistency was assessed post hoc using Cronbach’s alpha. Several dimensions fell below the conventional threshold of 0.60; these limitations have been explicitly addressed in the updated manuscript. We have emphasized the need for future studies to conduct formal pretesting and further validation. These revisions are reflected in Discussion (Lines 383-385, 395–397). |
||||||||||||||||||||||||||||||||||||||||||||||||||||||||||||||||||||||||||||||||||||||||||||||||||||||||||||||||||||||||||||||||||||||||||||
Comment 13 (Method): Feedback on Data Collection: ? Potential risk of bias in data collection: The method described (questionnaires delivered via various hierarchies — directors, committee heads, public health authorities) may introduce: Response bias due to institutional pressure: participants may feel compelled to respond in a certain way, knowing that the questionnaires pass through several hands before reaching the research team (despite being sealed). Risk of perceived breach of anonymity: It is not entirely clear how participants were assured that their responses would remain confidential. The authors are advised to comment on this potential bias and, if applicable, how it was mitigated (e.g., explicit communication about anonymity, how the process was explained to participants). |
||||||||||||||||||||||||||||||||||||||||||||||||||||||||||||||||||||||||||||||||||||||||||||||||||||||||||||||||||||||||||||||||||||||||||||
Response 13 (Method): Thank you for raising this concern. We agree that hierarchical distribution of questionnaires may introduce response bias. To address this, we clearly informed all participants, both verbally and in writing, that their responses would remain anonymous. As per HSOPSC guidelines, which advise against reporting facility-level results for hospitals with fewer than 10 respondents to protect anonymity, we also informed participants in district hospitals that individual hospital-level data would not be disclosed. These procedures are now described in the revised Methods (Lines 123–137) and acknowledged as a limitation in the Discussion (Lines 369–374). |
||||||||||||||||||||||||||||||||||||||||||||||||||||||||||||||||||||||||||||||||||||||||||||||||||||||||||||||||||||||||||||||||||||||||||||
Comment 14 (Method): Feedback on Analysis: ? Criteria for dealing with missing data The text states: “Participants with missing responses for any items within a dimension were excluded from these calculations.” This criterion is very restrictive and can lead to significant data loss. The authors should justify why they did not use alternative approaches (e.g., imputation, average of responses given) to avoid bias due to case exclusion. Indicate the proportion of data/exclusions per dimension due to missing data. |
||||||||||||||||||||||||||||||||||||||||||||||||||||||||||||||||||||||||||||||||||||||||||||||||||||||||||||||||||||||||||||||||||||||||||||
Response 14 (Method) Thank you for your detailed feedback on the analysis. We have revised the Methods and Results sections accordingly. Regarding missing data, we excluded participants with missing responses for any item within a given dimension to preserve the integrity of composite scores. Each composite measure contains only three to four items, and missing responses for any item within a given dimension substantially impact PRR accuracy. Given the small sample size and potential variability introduced by imputation, we chose not to apply substitution methods. However, these participants were retained in calculations for other dimensions to maintain the sample size. Exclusion rates per dimension ranged from 2.4% to 8.3%, except for “Communication Openness,” which had a rate of 16.2%. These details are reported in the Results section (Lines 198–199). |
||||||||||||||||||||||||||||||||||||||||||||||||||||||||||||||||||||||||||||||||||||||||||||||||||||||||||||||||||||||||||||||||||||||||||||
Comment 15 (Lack of logistic regression): Method: Feedback on Analysis: ? Statistical analysis — lack of detail on analytical model Although the tests used are mentioned (chi-square/Fisher), it is necessary to clarify: Whether there was adjustment for potential confounding factors (e.g., the type of hospital may be associated with other variables such as size, resources). Why only bivariate tests were chosen and not multivariate analyses, given the objective of the study. The results section: ? The authors compare dimensions between types of hospitals and professions using bivariate tests (chi-square, Fisher). It would be methodologically more robust to apply multivariate analyses (e.g., logistic regression or generalised linear models) to adjust for potential confounding factors (e.g., age, gender, type of hospital, years of experience). The absence of this adjustment limits the causal interpretation and generalisation of the findings. The discussion section: In the discussion the authors compare types of hospitals and professions, but the discussion does not address the relevant limitation: the differences observed may be associated with uncontrolled factors (e.g., age, experience, previous training). It is suggested to discuss that the absence of multivariate analyses limits the strength of the conclusions and that future studies should adjust for confounding variables. |
||||||||||||||||||||||||||||||||||||||||||||||||||||||||||||||||||||||||||||||||||||||||||||||||||||||||||||||||||||||||||||||||||||||||||||
Response 15 (Lack of logistic regression): Thank you for your detailed feedback regarding the use of bivariate tests. In this study, we applied bivariate analyses (chi-square and Fisher’s exact tests) to compare proportions across hospital types and professions. We acknowledge that multivariate models such as logistic regression or generalized linear models would have provided stronger control for potential confounding factors (e.g., hospital size, staffing, workload, and participant demographics). However, we did not conduct multivariate analyses for the following reasons: First, the exploratory nature of the study and the limited sample size posed statistical constraints. Second, the HSOPSC instrument is designed to assess safety culture at the hospital level, and cross-profession comparisons may be influenced by facility-level factors such as human resources and workload. In Laos, discrepancies between officially designated bed numbers and actual operational capacity are common, and staffing often relies on temporary or volunteer workers with fluctuating availability. Adjusting for such factors would require multilevel contextual data that were beyond this study’s scope. We have explicitly acknowledged this limitation in the revised Discussion section (Lines 391–395) and added a recommendation for future research.
|
||||||||||||||||||||||||||||||||||||||||||||||||||||||||||||||||||||||||||||||||||||||||||||||||||||||||||||||||||||||||||||||||||||||||||||
Comment 16 (Internal validity): Method: Feedback on Analysis: Reliability assessment: The authors used Cronbach's alpha with a threshold of 0.60. However, it is necessary to present the values obtained by dimension (not just mention the threshold). It would be relevant to discuss the impact of low alphas on the interpretation of the results, in addition to stating that they were ‘interpreted with caution’.
The results section: ? The alpha values presented range from 0.33 to 0.7. Although the authors mention interpreting dimensions with alpha < 0.60 with caution, they fail to discuss the concrete impact of these low values on the reliability of the results presented (e.g., Management Support for Patient Safety with 0.33). This could affect the validity of the findings reported in these dimensions. The discussion section: Although the discussion refers to low Cronbach's alpha values (e.g., Management Support for Patient Safety 0.33), there is no critical analysis of what this implies for the interpretation of data from these dimensions. It is important to recognize that the results of these dimensions should be read with caution, as low internal consistency may compromise the validity of the conclusions. |
||||||||||||||||||||||||||||||||||||||||||||||||||||||||||||||||||||||||||||||||||||||||||||||||||||||||||||||||||||||||||||||||||||||||||||
Response 16 (Internal validity): Thank you for your detailed feedback on the analysis. For reliability, we present Cronbach’s alpha values for each dimension in Table 3, which ranged from 0.33 to 0.70. Dimensions with alpha values below the conventional threshold of 0.60 were interpreted with caution and compared with results from previous studies (Lines 168–172). We have included a cautionary note regarding “Management Support for Patient Safety” in the Discussion (Lines 317–321), outlined the implications of low internal consistency, and explicitly noted this limitation in the revised manuscript (Lines 383–391). |
||||||||||||||||||||||||||||||||||||||||||||||||||||||||||||||||||||||||||||||||||||||||||||||||||||||||||||||||||||||||||||||||||||||||||||
Comment 17 (Result): The results section: ? The participant flow shows that 253 (75.5%) were included in the final analysis. It would be relevant to report whether there were systematic differences between those included and those excluded (e.g., by type of hospital, profession) to assess the risk of selection bias. |
||||||||||||||||||||||||||||||||||||||||||||||||||||||||||||||||||||||||||||||||||||||||||||||||||||||||||||||||||||||||||||||||||||||||||||
Response 17 (Results): The effective response rates from each hospital ranged from 52% to 84%, with an average of 70% among the four provincial hospitals (PHs), and from 40% to 100%, with an average of 78% among the district hospitals (DHs). These figures were added on Lines 183–186. Although 13.7% of participants were excluded as they did not return the questionnaire, the anonymous design of the survey prevented us from comparing characteristics between returned and non-returned responses. The questionnaires were specifically distributed to members of patient safety committees, but we could not identify the respondents within each hospital. However, 20 participants were excluded for either declining participation or not signing the consent form; all 20 participants were from DHs. In three DHs, four questionnaires each were excluded for missing consent forms, which may have affected selection bias. Eight participants under the age of 19 were excluded as they were not the target participants; the study focused on safety committee members expected to be experienced and qualified. An additional eight were excluded due to missing age data. Their characteristics were similar to those of the included participants, except for a higher proportion of missing information These participants constituted a small proportion of the total participants. The following table presents this comparison. [Table] Characteristic comparison between included participants and dropped participants due to unknown age.
Given the small number of these participants, we chose not to include them in the manuscript and acknowledged the potential selection bias in the Discussion (Lines 366–369). |
||||||||||||||||||||||||||||||||||||||||||||||||||||||||||||||||||||||||||||||||||||||||||||||||||||||||||||||||||||||||||||||||||||||||||||
Comment 18 (Result): The results section: ? The results are mainly descriptive (percentages by dimension and groups), but, a more in-depth analysis of the magnitude and practical relevance of the observed differences is lacking (e.g., what does a difference of ~12% in Supervisor/Manager Expectations between PHs and DHs mean in practice?). Furthermore, there is no discussion of whether these differences are consistent with previous studies or with HSOPSC benchmarks. |
||||||||||||||||||||||||||||||||||||||||||||||||||||||||||||||||||||||||||||||||||||||||||||||||||||||||||||||||||||||||||||||||||||||||||||
Response 18 (Result): Thank you for this valuable comment. In response, we have added interpretation based on HSOPSC benchmarks and discussed the magnitude and practical implications of observed differences, including the approximately 12% difference in “Supervisor/Manager Expectations and Actions Promoting Patient Safety” between PHs and DHs. We noted that higher PH scores may reflect stronger institutional structures and more established supervisory mechanisms. Additionally, we compared our findings with results from other HSOPSC studies (Table 5) and discussed whether the observed patterns align with or diverge from those benchmarks. These additions appear in the revised Discussion section on Lines 284-296. |
||||||||||||||||||||||||||||||||||||||||||||||||||||||||||||||||||||||||||||||||||||||||||||||||||||||||||||||||||||||||||||||||||||||||||||
Comment 19 (Result): The results section: ? It is reported that less than a third of participants had patient safety training. The relationship between this variable and perceptions of safety culture could be explored, given its potential explanatory role. |
||||||||||||||||||||||||||||||||||||||||||||||||||||||||||||||||||||||||||||||||||||||||||||||||||||||||||||||||||||||||||||||||||||||||||||
Response 19 (Result): Thank you for your important feedback on the results. In our study, fewer than one-third of the participants reported receiving patient safety training. However, during qualitative interviews at the target hospitals, we observed that the term “patient safety training” was interpreted inconsistently. Some participants referred to administrative or hand hygiene training, indicating varied understandings of the concept. This makes comparison difficult without an additional definition in the questionnaire. Furthermore, a preliminary comparison of PRRs between participants with and without self-reported training did not yield statistically significant differences. As a result, we did not perform further analysis. We have added this explanation and its implications to the revised Results section on Lines 194–196. |
||||||||||||||||||||||||||||||||||||||||||||||||||||||||||||||||||||||||||||||||||||||||||||||||||||||||||||||||||||||||||||||||||||||||||||
Comment 20 (Discussion): The discussion section: ?The authors state that the results may apply to other regions of Laos. However, as they acknowledge in the limitations section, the study only included hospitals in the south and safety committee professionals. ?Although leadership and a non-punitive culture are identified as necessary, more operational suggestions are lacking (e.g., specific training, strengthening adverse event reporting systems, benchmarking initiatives). |
||||||||||||||||||||||||||||||||||||||||||||||||||||||||||||||||||||||||||||||||||||||||||||||||||||||||||||||||||||||||||||||||||||||||||||
Response 20 (Discussion): Thank you for your thoughtful comment; we agree with the reviewer’s observations. Regarding generalizability, all four provinces in this study had established patient safety committees at the time of data collection, possibly indicating a more developed safety culture compared with other regions in Laos. However, as Laos is implementing national quality standards across all healthcare facilities, including incident reporting systems, the insights from these provinces may become increasingly applicable to other provinces as similar systems are adopted nationwide. We have also clarified this in the revised Discussion section (Lines 354–360) and noted in the Introduction (Lines 61–69) that public hospitals are the main care providers. As for operational recommendations, we added practical examples highlighting the importance of structured incident reporting systems and targeted training for nurse managers. Specifically, we recommend Just Culture training focused on human risk factors, system-level prevention strategies, and learning from incidents rather than assigning blame. These suggestions were based on prior studies on safety culture interventions and are included in the revised Discussion (Lines 324–338). After this survey, we also implemented such training in the four provinces and hope to share these outcomes in future work. |
||||||||||||||||||||||||||||||||||||||||||||||||||||||||||||||||||||||||||||||||||||||||||||||||||||||||||||||||||||||||||||||||||||||||||||
Comment 21 (Conclusion): The conclusion is underdeveloped; further development of this section is recommended. The final paragraph does not sufficiently reflect the methodological limitations identified (e.g., low reliability of some dimensions, sample restricted to committee members, lack of adjustment for confounders). This could lead to an overestimation of the conclusions. The conclusions should be moderate and acknowledge the limitations that condition the generalization of the findings. The text mentions strengthening leadership to foster a non-punitive culture, but does not translate these recommendations into concrete practical actions (e.g., training, review of incident reporting policies, creation of safe reporting channels). The authors could make the recommendations more applicable and specific to the context studied. When stating that the results can improve the quality of healthcare, there is a lack of direct evidence in the study to support this association (quality indicators were not measured). They can be reformulated to reflect that these are hypotheses that should be tested in future interventions. |
||||||||||||||||||||||||||||||||||||||||||||||||||||||||||||||||||||||||||||||||||||||||||||||||||||||||||||||||||||||||||||||||||||||||||||
Response 21 (Conclusion): Thank you for your insightful comments. In response, we have revised the Conclusion section to
These revisions are reflected in the revised Conclusion section (Lines 399–412). |
||||||||||||||||||||||||||||||||||||||||||||||||||||||||||||||||||||||||||||||||||||||||||||||||||||||||||||||||||||||||||||||||||||||||||||
4. Response to Comments on the Quality of English Language |
||||||||||||||||||||||||||||||||||||||||||||||||||||||||||||||||||||||||||||||||||||||||||||||||||||||||||||||||||||||||||||||||||||||||||||
Thank you for your comment. We have carefully reviewed and shortened lengthy sentences. |

Reviewer 2 Report
Comments and Suggestions for Authors
Dear authors:
I have reviewed your paper titled "Patient Safety Culture of Hospitals in Southern Laos: A Cross-sectional Study with the Hospital Survey on Patient Safety Culture". This cross-sectional study was conducted across 27 hospitals in southern Laos. The results were analyzed from 253 participants responses to the Hospital Survey on Patient Safety Culture, covering 12 dimensions. The results suggested strengths in teamwork and organizational learning while identifying nonpunitive responses and event reporting as critical areas for improvement in hospitals in Laos.
I congratulate you on your choice of topic—this is a highly relevant and important health management area of research that underscores the need for healthcare quality and protecting patients and health professionals.
I also want to acknowledge the methodological rigor, and ethical standards adhered to in your study.
As an external reviewer, I would like to offer a few suggestions that may further enhance the clarity and completeness of your manuscript:
ABSTRACT: please insert the objective of your study, it is not clear.
Introduction: please uniformize the concept: healthcare professionals (line 16; 278) or healthcare workers (line 37; 85); Could be improved, justify the importance to health systems, health institutions, patients and community and healthcare professionals.
Method: Instrument:
1) Please let us know in the process of translation, did you contact the original authors, to obtain authorization and to validate your translation purpose? 2) did you validate the content? If yes, how did you make it? 3) Once you calculate internal consistency, alpha de Cronbach value could be an interesting insert that results here in instrument characterization.
Discussion: In discussion it is not expected to have tables that should be in results or methods. (table 5 is results should be in that part).
Limitations: the internal consistency of some dimensions/factors are very low.
Conclusion: this could be improved, highlight the implications and contributions that the study brings; and could give some clues to future research.
I have no additional suggestions at this time and would like to extend my best wishes for the publication of your paper. I thoroughly enjoyed reading it.
Best regards,
Comments on the Quality of English LanguageThe English language did not hinder my understanding of the document.
Author Response
Comments 1: ABSTRACT: please insert the objective of your study, it is not clear. Introduction: please uniformize the concept: healthcare professionals (line 16; 278) or healthcare workers (line 37; 85); Could be improved, justify the importance to health systems, health institutions, patients and community and healthcare professionals. |
Response 1: Thank you for your insightful comments. We have revised the Abstract to clearly state the research objective (Lines 15–17). In the Introduction, we have standardized terminology by consistently using the term “healthcare workers” throughout. Additionally, we have expanded the explanation in the Introduction to further justify the relevance of patient safety culture for health systems, institutions, patients, and healthcare workers (Lines 41–45). |
Comments 2: Method: Instrument: 1) Please let us know in the process of translation, did you contact the original authors, to obtain authorization and to validate your translation purpose? 2) did you validate the content? If yes, how did you make it? 3) Once you calculate internal consistency, alpha de Cronbach value could be an interesting insert that results here in instrument characterization. |
Response 2: Thank you for sharing these valuable concerns regarding the instrument.
|
Comments 3: Discussion: In discussion it is not expected to have tables that should be in results or methods. (table 5 is results should be in that part). |
Response 3: Thank you for this helpful comment. We understand that tables are generally expected in the Results or Methods sections. However, Table 5 does not present original findings from this study. Instead, it supports the Discussion by comparing our results with previous HSOPSC-based studies in other countries. As this comparison serves as a point of reflection rather than formal analysis, we believe its inclusion in the Discussion is appropriate. Furthermore, we used a table format to present the data to enhance clarity and facilitate cross-country comparison. |
Comments 4: Limitations: the internal consistency of some dimensions/factors are very low. |
Response 4: Thank you for your comment. We agree with your suggestion and have added the above point to the limitations section (Lines 383–385). We also explained this limitation in the conclusion section. |
Comments 5: Conclusion: this could be improved, highlight the implications and contributions that the study brings, and could give some clues to future research. |
Response 5: Thank you for this suggestion. In response, we have revised the Conclusion to better emphasize the study’s implications and contributions. We have now highlighted how the findings provide critical baseline data on patient safety culture in Laos, which may guide future safety initiatives. We have also outlined directions for future research, including the need for multivariate analyses, further validation of the adapted tool, and qualitative studies to explore underlying contextual factors. These revisions are included in the updated Conclusion section (Lines 398-412). |
4. Response to Comments on the Quality of English Language |
Thank you for your comment. We have carefully reviewed the language for resubmission to ensure clarity and precision. |
Reviewer 3 Report
Comments and Suggestions for Authors
It is my pleasure to review the manuscript entitled “Patient Safety Culture of Hospitals in Southern Laos: A Cross-sectional Study with the Hospital Survey on Patient Safety Culture.” In this article, the authors conducted a cross-sectional study across 27 hospitals in southern Laos using the Hospital Survey on Patient Safety Culture (HSOPSC). The study provides timely and essential insights into patient safety culture (PSC) in a low-resource setting where such evaluations have been limited. Understanding the strengths and weaknesses in PSC is vital for guiding quality improvement in healthcare services in Laos.
- Abstract: The abstract presents a general overview of the aim, methodology, and key findings. Need summarizes the statistical approach and lacks specific details about the sample characteristics and key statistical findings. The abstract could benefit from explicitly mentioning the sample size, sampling method.
- Introduction: The research problem is implied rather than explicitly stated; however, the research problem, limited research on PSC in Laos. The problem statement is present but implicit. The authors highlight the absence of standardized PSC research in Laos and poor safety performance in district hospitals as the justification for the study. However, it would benefit from being more formally presented. The introduction lacks critical synthesis of the literature—studies are cited in sequence rather than analyzed comparatively.
- Setting: need rational and how select setting.
- Sample: need specific Inclusion Criteria of experience, and rational of these Inclusion Criteria.
- Instrument: need specific point Likert scale and a range of score and interpretation and if there is cut point, also need internal reliability. How to calculate each total instrument and subscales. Also, need to explore if there are reverse items.
- Data Collection: need describe how to collect (paper or online) and who distribute Questionnaires and how.
- Ethical Considerations: need to mention if you get permission to use and translate and if the main author revised the tool.
- Results: need to report on the level of each scale. about 3.3. Levels of Conflict Resolution Styles it is report regression not level, so please write appropriate subheading.
- Discussion: The discussion fails to explore why teamwork and organizational learning are rated high in a healthcare context known to lack structural safety systems or standardized safety protocols. Likewise, the emphasis on a “blame culture” as an explanation for poor reporting is conceptually valid but is not supported by qualitative or contextual evidence within this study. The authors do not sufficiently explain why some dimensions scored high (e.g., “Organizational Learning”), especially given the lack of supporting policies, infrastructure, or prior PSC development in Laos. The discussion omits potential language or literacy barriers, particularly given that one HSOPSC item had to be excluded due to mistranslation.
- The discussion fails to explore the implications of missing demographic data (e.g., gender, years of experience) on subgroup results. No specific strategies are proposed, such as: Integrating PSC into pre-service nursing and medical curricula. Developing feedback mechanisms or whistleblower protection policies. Leveraging community-based participatory research to address rural hospital deficits.
- Similarly, research implications are limited to suggesting periodic reassessments, without addressing: Psychometric revalidation of HSOPSC in Lao PDR. Development of context-specific instruments. Need for qualitative research to explore why errors go unreported.
Author Response
Comments 1: Abstract: The abstract presents a general overview of the aim, methodology, and key findings. Need summarizes the statistical approach and lacks specific details about the sample characteristics and key statistical findings. The abstract could benefit from explicitly mentioning the sample size, sampling method. |
Response 1: Thank you for the helpful suggestion. We have revised the Abstract to include the sampling method, sample size, and a summary of the statistical approach, including descriptive statistics and bivariate analysis (chi-square and Fisher’s exact tests). These changes appear on Lines 17–22 of the updated manuscript. |
Comments 2: Introduction: The research problem is implied rather than explicitly stated; however, the research problem, limited research on PSC in Laos. The problem statement is present but implicit. The authors highlight the absence of standardized PSC research in Laos and poor safety performance in district hospitals as the justification for the study. However, it would benefit from being more formally presented. The introduction lacks critical synthesis of the literature—studies are cited in sequence rather than analyzed comparatively. |
Response 2: Thank you for the constructive feedback. We agree that the original Introduction lacked an explicit research problem statement and critical synthesis of the existing literature. Accordingly, we have revised the Introduction (Lines 54–58) to more clearly articulate the literature gap by noting that although patient safety culture has been studied in other Southeast Asian countries, no prior study has applied the HSOPSC tool in Laos. We also strengthened the justification for this study by emphasizing the importance of understanding local patient safety culture in informing context-specific interventions. Additionally, we clarified the study aim in a separate statement (Lines 77–78), improving both the clarity of the research problem and the rationale for conducting this study. |
Comment 3 (Method): • Setting: need rational and how select setting. • Sample: need specific Inclusion Criteria of experience, and rational of these Inclusion Criteria. • Instrument: need specific point Likert scale and a range of score and interpretation and if there is cut point, also need internal reliability. How to calculate each total instrument and subscales. Also, need to explore if there are reverse items. • Data Collection: need describe how to collect (paper or online) and who distribute Questionnaires and how. • Ethical Considerations: need to mention if you get permission to use and translate and if the main author revised the tool. |
Response 3 (Method): Thank you for your thorough comments on the Methods section. We have revised and expanded this section to improve clarity and transparency as follows:
These revisions improve methodological clarity and replicability. |
Comment 4: Results: need to report on the level of each scale. about 3.3. Levels of Conflict Resolution Styles it is report regression not level, so please write appropriate subheading. |
Response 4: Thank you for your comment. We believe there may have been a misunderstanding regarding the subheading “3.3. Levels of Conflict Resolution Styles” as this phrase does not appear in our manuscript. It may have been carried over from another article or template. In our Results section, we primarily report the percentage of positive responses for each HSOPSC dimension and conduct group comparisons using chi-square or Fisher’s exact tests. We also report Cronbach’s alpha values for internal consistency and identify dimensions with high or low perceived patient safety culture. These are descriptive results and not based on regression analysis of conflict resolution styles. |
Comment 5: Discussion: The discussion fails to explore why teamwork and organizational learning are rated high in a healthcare context known to lack structural safety systems or standardized safety protocols. Likewise, the emphasis on a “blame culture” as an explanation for poor reporting is conceptually valid but is not supported by qualitative or contextual evidence within this study. The authors do not sufficiently explain why some dimensions scored high (e.g., “Organizational Learning”), especially given the lack of supporting policies, infrastructure, or prior PSC development in Laos. |
Response 5: Thank you for your thoughtful and important comment. We agree that the original Discussion lacked sufficient context regarding high ratings for dimensions such as “Organizational Learning—Continuous Improvement” despite limited national-level strategies in Laos. To address this, we have added a contextual explanation to the revised Discussion section. Specifically, we noted that continuous quality improvement efforts in the four southern provinces were supported by JICA’s technical cooperation and aligned with national quality standards. Hospitals in Laos are required by the Ministry of Health to form healthcare quality committees and develop action plans based on quality assessments, as part of the national hospital accreditation process. These factors may have contributed to higher scores in organizational learning (Lines 240-243, 358–360). For the dimension “Nonpunitive Response to Error,” we have clarified that similarly low scores have been reported in HSOPSC studies in other countries (Table 5). We have acknowledged the need for additional qualitative or contextual research to better understand cultural barriers to reporting and included this as a recommendation for future research (Lines 410–412). We have also added an interpretation of the most notable differences between provincial and district hospitals for selected dimensions such as “Supervisor/Manager Expectations” and discussed these in the context of the healthcare system in Laos (Lines 284–292). |
Comment 6: • The discussion omits potential language or literacy barriers, particularly given that one HSOPSC item had to be excluded due to mistranslation. • No specific strategies are proposed, such as: Integrating PSC into pre-service nursing and medical curricula. Developing feedback mechanisms or whistleblower protection policies. Leveraging community-based participatory research to address rural hospital deficits. • Similarly, research implications are limited to suggesting periodic reassessments, without addressing: Psychometric revalidation of HSOPSC in Lao PDR. Development of context-specific instruments. Need for qualitative research to explore why errors go unreported. • The discussion fails to explore the implications of missing demographic data (e.g., gender, years of experience) on subgroup results. |
Response 6: Thank you for your comments. We agree with your suggestions and have revised the manuscript accordingly. • First, we have explained in detail for the excluded item in the Methods section, “one item was excluded due to a misalignment in the questionnaire version during adaptation” and acknowledged this exclusion as a limitation (Lines 151–158, 380–383). • Second, we have added concrete recommendations informed by our findings (Lines 324–338). We also included concrete recommendations in the conclusion. • Third, we have included recommendations for further research, emphasizing the need for qualitative and multivariate approaches (Lines 395–397, 410-412). • Fourth, we have acknowledged that the absence of demographic variables such as gender and years of experience limits subgroup analysis and may introduce uncontrolled confounding. Owing to the exploratory nature of the study and limited sample size, multivariate analysis was not conducted. Adjusting for hospital-level contextual factors (e.g., staffing, workload) was also not feasible using the current dataset. We have acknowledged this limitation and recommended that future research incorporate more detailed demographic and institutional data to enable more robust comparisons (Lines 391–397). |
4. Response to Comments on the Quality of English Language |
Thank you for noting that English clarity could be improved. We have carefully reviewed the manuscript and submitted the revised version to a professional English-editing service. |
Reviewer 4 Report
Comments and Suggestions for Authors
-
If available, citing studies that successfully used HSOPSC in countries with similar contexts (low-resource or Southeast Asian nations) could enhance justification.
-
A brief note about HSOPSC's reliability/validity or limitations (if applicable) might provide a more balanced evaluation.
Author Response
3. Point-by-point response to Comments and Suggestions for Authors |
Comments 1: If available, citing studies that successfully used HSOPSC in countries with similar contexts (low-resource or Southeast Asian nations) could enhance justification. |
Response 1: Thank you for pointing this out. We agree with this comment. We cited the studies in Vietnam in the discussion part to enhance the justification of HSOPSC in Line 66-68 of the updated version with track changes. |
Comments 2: A brief note about HSOPSC's reliability/validity or limitations (if applicable) might provide a more balanced evaluation. |
Response 2: Thank you for pointing this out. We agree with this comment. We briefly mentioned the reliability/validity of HSOPSC in the introduction session on Lines 55-71, and the limitation of HSOPSC that some dimensions have challenges in reaching a desirable level of reliability on Lines 221-225 of the updated version with track changes. |
4. Response to Comments on the Quality of English Language |
Thank you for mentioning that English is fine and does not require any improvement. We carefully reviewed English for resubmission. |

Round 2
Reviewer 1 Report
Comments and Suggestions for Authors
The manuscript's structure, clarity, and methodological transparency have all been enhanced by the authors. Although some interpretations are limited by the lack of multivariate analyses, this is sufficiently acknowledged in the discussion. The manuscript satisfies publication requirements and offers insightful information about patient safety culture. These dimensions could be further investigated in future research using larger samples and more intricate models.